# Low-Volume Metastases in Cervical Cancer: Does Size Matter?

**DOI:** 10.3390/cancers16061107

**Published:** 2024-03-09

**Authors:** Tommaso Bianchi, Tommaso Grassi, Giampaolo Di Martino, Serena Negri, Gaetano Trezzi, Robert Fruscio, Fabio Landoni

**Affiliations:** 1Department of Medicine and Surgery, University of Milano-Bicocca, 20126 Milano, Italy; t.bianchi4@campus.unimib.it (T.B.); robert.fruscio@unimib.it (R.F.); fabio.landoni@unimib.it (F.L.); 2Clinic of Obstetrics and Gynecology, IRCCS Fondazione San Gerardo dei Tintori, 20900 Monza, Italygaetano.trezzi@gmail.com (G.T.)

**Keywords:** cervical cancer, low-volume metastases, micrometastases, isolated tumor cells, ultrastaging, sentinel lymph node

## Abstract

**Simple Summary:**

The reported incidence of low-volume metastases (LVM) in early-stage cervical cancer ranges from 4 to 20%. Given the conflicting results of prospective and retrospective studies, their prognostic value is still debated, especially for isolated tumor cells (ITC). This narrative review aims to highlight current evidence, controversies, and unanswered questions about the definition and prognostic role of LVM.

**Abstract:**

The implementation of sentinel lymph node (SLN) biopsy is changing the scenario in the surgical treatment of early-stage cervical cancer, and the oncologic safety of replacing bilateral pelvic lymphadenectomy with SLN biopsy is currently under investigation. Part of the undisputed value of SLN biopsy is its diagnostic accuracy in detecting low-volume metastases (LVM) via pathologic ultrastaging. In early-stage cervical cancer, the reported incidence of LVM ranges from 4 to 20%. The prognostic impact and the role of adjuvant treatment in patients with LVM is still unclear. Some non-prespecified analyses in prospective studies showed no impact on the oncologic outcomes compared to node-negative disease. However, the heterogeneity of the studies, the differences in the disease stage and the use of adjuvant treatment, and the concomitant pelvic lymphadenectomy (PLND) make reaching any conclusions on this topic hard. Current guidelines suggest considering micrometastases (MIC) as a node-positive disease, while considering isolated tumor cells (ITC) as a node-negative disease with a low level of evidence. This review aims to highlight the unanswered questions about the definition, identification, and prognostic and therapeutic roles of LVM and to underline the present and future challenges we are facing. We hope that this review will guide further research, giving robust evidence on LVM and their impacts on clinical practice.

## 1. Introduction

Radical hysterectomy (RH) with sentinel lymph node (SLN) biopsy and bilateral pelvic lymph node dissection (PLND) represents the standard treatment for women with early-stage cervical cancer and no child-bearing desire [1]. Pelvic lymph node involvement is the major prognostic factor in patients with early-stage cervical cancer [2], and an adequate assessment of pelvic lymph node status is mandatory to guide the most appropriate treatment choice. In the last two decades, SLN biopsy has been progressively implemented in routine clinical practice for early-stage cervical cancer treatment. As for other malignancies (i.e., breast cancer and melanoma), the SLN represents the first node draining the lymphatic vessels from the primary cancer. In cervical cancer, its status reflects the status of all other pelvic (and para-aortic) lymph nodes. Two prospective French multicenter clinical trials (SENTICOL 1 [3] and SENTICOL 2 [4]) demonstrated the feasibility, diagnostic accuracy, and safety of SLN biopsy in early-stage cervical cancer. Two further prospective clinical trials (SENTIX [5] and SENTICOL 3 [6]) are currently ongoing. They will probably give the final answer as to whether SLN biopsy alone will replace bilateral PLND in the surgical treatment of early-stage cervical cancer. Low-volume metastases (LVM) are a peculiar pathologic finding of SLN biopsy. Lymph node involvement is categorized as having macrometastases (MAC), micrometastases (MIC), or isolated tumor cells (ITC) depending on the size of the nodal metastasis. Metastases larger than 2 mm in size are classified as MAC; metastases between 0.2 mm and 2 mm in size and/or comprising more than 200 tumor cells are classified as MIC; and ITC indicates tumor deposits comprising less than 200 cells and/or less than 0.2 mm in size [7]. MIC and ITC are both included in the LVM definition. The literature still debates the clinical and prognostic impacts of LVM. This critical review points out the open questions of the available literature on this topic. We hope to stimulate further research to give robust evidence about the oncological implications of LVM.

## 2. Sentinel Lymph node Assessment: How Low-Volume Metastases Are Detected

SLN biopsy has two significant diagnostic advantages compared to bilateral PLND. First, SLN mapping can lead to the identification of metastatic lymph nodes in atypical lymphatic drainage pathways. In the SENTIX trial, 4% of patients had their SLN detected in the common iliac and pre-sacral areas, which are seldom included in PLND [8]. Second, SLN biopsy is more accurate than PLND in assessing pelvic nodal involvement. SLNs are indeed submitted to an extensive pathological analysis called “ultrastaging”. It consists of cutting serial sections of each SLN paraffin block, on which additional hematoxylin and eosin (H&E) stains and immunohistochemistry (IHC) are performed (Figure 1). Ultrastaging is performed if the SLN is negative for metastases at the first routine H&E section. This extensive pathological staging increases the detection of nodal metastases due to the multiple levels of examination of the SLNs [9]. Mainly, it increases the probability of detecting LVM, such as MIC and ITC, which are usually missed when conventional H&E analysis is performed. In a retrospective series by Cibula et al. [10], traditional pathologic evaluation of SLN would have failed to identify 11% of patients with nodal involvement due to LVM recognizable only with ultrastaging.

## 3. Are Low-Volume Metastases Clinically Meaningful? Controversies of the Current Evidence

Among the literature’s data on the clinical impact of LVM on recurrence and survival in early-stage cervical cancer, the results of the two prospective trials SENTICOL 1 and SENTICOL 2 conflict with most of the retrospective series available (Table 1).

SENTICOL 1 was a French prospective longitudinal study that evaluated the feasibility and diagnostic accuracy of SLN biopsy in the treatment of patients with FIGO IA1 with lymph-vascular space invasion (LVSI) to IB1 cervical carcinoma [3]. All patients received SLN biopsy and subsequent PLND, irrespective of the results of intraoperative frozen section analysis of the SLN biopsies. Conversely, SENTICOL 2 was a prospective multicenter randomized trial assessing the morbidity and oncologic safety of SLN biopsy alone compared to SLN biopsy followed by PLND in patients with 2009 FIGO IA1 with LVSI, IA2, IB1, and IIA1 cervical carcinoma [4]. In 2020, Guani et al. performed a conjunct analysis of the results of both trials, focusing on the impact of LVM on disease-free survival (DFS) [11]. Among the 321 SENTICOL 1 and 2 patients, 24 (7.5%) had LVM. In particular, MIC were found in 11 patients (3.4%) and ITC in 13 patients (4.1%). No statistically significant difference in 3 years-DFS was found between patients with LVM and node-negative patients (92.7% vs. 93.6%, respectively). Two retrospective series align with these results [12,13]. Stany et al. [12] performed a retrospective review of 129 patients treated for early-stage cervical cancer. The ultrastaging of pelvic nodes identified 26 patients with pelvic LVM previously considered node-negative at routine H&E examination. No differences in risk of recurrence or death were found in patients with LVM compared with node-negative patients. Similarly, in their multicenter retrospective case–control study, Buda et al. [13] evaluated the impact of LVM on DFS in 573 women with FIGO 2018 IA2-IB2 cervical cancer. In all of the centers, the same ultrastaging protocol was adopted in the pathologic processing of SLNs [14]. Among the 573 patients included in the study, 85 (15%) had positive nodes, and 21 (3.6%) were found with MIC or ITC only in the SLN. No differences in risk of relapse or DFS were found in the LVM population (OR 0.65; 95% CI 0.36–1.20) compared to the node-negative group.

These findings contrast with the results of a large retrospective study by Cibula et al. and other retrospective series [15,16,17,18,19]. In 2012, Cibula et al. [15] collected the largest retrospective series addressing the prognostic significance of LVM in early-stage cervical cancer. Six hundred and forty-five patients with FIGO 2009 IA2-IIB cervical carcinoma, from eight tertiary centers, were included in the analysis. All patients received an SLN biopsy followed by systematic bilateral PLND, irrespective of the uterine procedure performed. MIC and ITC were found in 7.1% and 3.9% of patients, respectively. The presence of MIC was a significant independent factor for reduced overall survival (OS) (HR 4.60, 95% CI 1.34–15.77). Conversely, ITC did not increase the risk of recurrence, nor were they associated with decreased OS. Similar results were found in the single-institution series reported by Kocian et al. [16]. Among the 226 patients with early-stage cervical carcinoma, 14 (6.2%), 16 (7.1%), and 8 (3.5%) patients were found with MAC, MIC, and ITC in the SLN, respectively. Of note, two patients with MIC and one with ITC in the SLN also had MIC and ITC in the non-sentinel pelvic lymph nodes, respectively. After a median follow-up of 65 months, DFS reached 93%, 89%, 69%, and 87% in the node-negative, MAC, MIC, and ITC groups, respectively.

A recently published meta-analysis by Guani et al. aimed to comprehensively evaluate the available data on the clinical impact of LVM in early-stage cervical cancer [20]. Data for a total of 2191 patients from seven studies were retrieved. The survival analyses compared the LVM (MIC + ITC) population to the node-negative one. Comprehensively, negative impacts on DFS and OS were found for LVM, with HRs of 2.60 (95% CI: 1.55–4.34) and 5.65 (95% CI: 2.81–11.39), respectively.

Despite the undoubted robustness of Guani et al.’s metanalysis findings, some questions remain unresolved. First, the definition of LVM is arbitrary and adapted from breast cancer staging [7]. Second, no specific analysis was performed for ITC due to the exiguity of the ITC population and the limited number of events. Therefore, no conclusions could be drawn on their clinical significance. Third, the studies included in the metanalysis are limited by a great heterogeneity concerning the study design, the sample size, the stage of disease included in the study, the use of adjuvant treatment, and the concomitant PLND and surgical approach. For example, in the Cibula et al. series [15], even patients with 2009 FIGO stage IB2, IIA1-2, and IIB cervical cancer were included. These tumors are now almost universally considered “locally advanced” and treated with exclusive curative radio-chemotherapy. Tumor size and upper vaginal and parametrial involvement are well-known risk factors for disease relapse and nodal involvement. Including patients with independent poor prognostic factors could have hidden the real impacts of LVM on recurrence and survival. Fourth, the lack of a shared ultrastaging technique and protocol may impact the detection of LVM and, therefore, could have influenced the results. An internationally validated protocol for SLN ultrastaging has not been proposed yet, and ultrastaging techniques vary among different institutions. Given the importance of ultrastaging in the detection of LVM, the lack of a standardized protocol raises the question of whether the literature’s data on the prognostic impacts of LVM are comparable. Last of all, little information is available on the recurrence time and site in cases of primary treatment failure. In their meta-analysis, Guani et al. [20] estimated that most recurrences in patients with MIC occurred within 3 years after primary treatment. However, they could not retrieve any information about the recurrence site or draw any conclusions on the impact of LVM on disease diffusion and recurrence patterns.

### 3.1. Macrometastases, Micrometastases, and Isolated Tumor Cells: A Matter of Size?

Lymph node involvement is categorized as MAC, MIC, or ITC depending on the size of the nodal metastasis according to the American Joint Committee on Cancer (AJCC) recommendations on breast cancer staging [7]. It has been recently implemented in the TNM 8, where MAC and MIC are reported as pN1 and ITC as pN0. Subsequently, these categories have been widely applied to other tumor sites and gynecological cancers (i.e., endometrial and vulvar). Still, the clinical impacts of these different types of metastases have not yet been validated specifically for cervical cancer.

Recently, Dostalek et al. [21] addressed this issue and performed a subgroup analysis of the SCCAN project, a multicenter retrospective observational cohort study evaluating the recurrence patterns in patients with cervical cancer. In their series of 172 patients with nodal involvement (79 MAC, 54 MIC, and 39 ITC), the authors tried to select the minimum cut-off of nodal metastasis size associated with a better prognosis compared to the entire cohort of node-positive patients. Particularly, they rearranged the cohort with LVM by dividing patients into subgroups according to the size of metastases at intervals of 0.1 mm, ranging from 0.1 to 1 mm. They found that patients with more than 0.4 mm nodal metastases had significantly shorter DFS than those of node-negative patients (HR 2.3; 95% CI 1.2–4.6). Additionally, they could not find a metastasis size cut-off that was able to identify a subgroup of patients with a significantly better prognosis than the node-positive cohort. Therefore, the authors concluded that stratifying patients into MIC and MAC according to the size of nodal metastases is artificial and has no prognostic impact. They suggested that patients with positive nodes should be managed uniformly until further evidence regarding the prognostic impact of “very low” volume metastases is available.

Comprehensively studied, MIC seems to be associated with a poorer prognosis compared to ITC. As previously described, in Guani et al.’s meta-analysis [20], the authors performed a survival analysis in two different settings, separately comparing the LVM (MIC + ITC) and the MIC populations to the node-negative cohort. Higher risks of recurrence and death were found in the MIC population compared to the combined LVM group. Particularly, HRs for recurrence were 2.60 (95% CI: 1.55–4.34) and 4.10 (95% CI: 2.71–6.20) in the LVM and MIC population, respectively, whereas HRs for death reached 5.65 (95% CI: 2.81–11.39) and 6.94 (95% CI: 2.56–18.81) in the corresponding subgroups. Although a specific analysis for ITC was not performed due to the limited numbers of cases and events, the lower hazards in the combined LVM population suggest that patients with ITC had a better prognosis than the MIC population.

### 3.2. Impacts of Adjuvant Therapy and Complementary PLND

Pelvic lymph node involvement is considered a high-risk factor for treatment failure, like parametrial invasion and positive surgical margins. These factors represent an indication for adjuvant radio-chemotherapy [1]. Involvement of pelvic (and para-aortic) lymph nodes has historically been detected by standard routine pathologic examination with H&E staining, and only in the last two decades has the issue of LVM emerged. A consensus on the indication for adjuvant treatment in the case of LVM is currently lacking. The 2018 FIGO staging [22] includes MIC in the node-positive group, while it is suggested that the presence of ITC is recorded without changing the stage. Discordance in treatment indications and patient selection emerge in prospective and retrospective studies addressing the clinical impact of LVM [11,16,19]. Despite this, clinical practice has changed over the years, mainly due to the findings of early retrospective reports that elucidated the possible detrimental effect of LVM on survival [15,16,17,18,19].

Notably, pelvic lymph node involvement, either macro- or micrometastatic, is usually associated with local tumor risk factors, such as LVSI, deep stromal invasion and tumor size. The combination of these tumor risk factors has historically represented an indication for adjuvant radiotherapy according to Sedlis’ criteria (intermediate-risk group) [23]. Therefore, in many studies evaluating the impact of LVM on oncologic outcomes in patients with early-stage cervical cancer, a non-negligible amount of patients received adjuvant treatment irrespective of LVM nodal status (Figure 2).

Similar to historical studies evaluating the extent of parametrial radicality in the surgical treatment of early-stage cervical cancer [25], the use of adjuvant therapy may have interfered with the real impact of LVM on prognosis. For example, in SENTICOL 1 [3], patients with LVM received adjuvant treatment if other concomitant risk factors were present. Of the 13 patients with MIC or ITC, 4 (31%) received an adjuvant treatment. In particular, one received adjuvant radio-chemotherapy because of parametrial involvement, and the other three received adjuvant radiotherapy in the presence of LVSI. Similarly, in the retrospective series by Stany et al. [12], no differences in risks of recurrence or death were found in patients with LVM compared with node-negative patients, but the post-operative radiation rate was significantly higher in the LVM group (38.5% vs. 18.4%). On the one hand, this substantiates the association of the presence of LVM with other significant prognostic factors that are likely to be an indication for adjuvant treatment. On the other hand, the use of adjuvant radiotherapy could have masked the detrimental effect of LVM on survival outcomes. 

Contrarily, in the already mentioned series of Cibula et al. [15], the presence of LVM was associated with decreased OS, even if a high proportion of patients with MIC and ITC received adjuvant treatment. Overall, 33% of patients received adjuvant radiotherapy or radio-chemotherapy, whereas in the LVM population 82.6% of patients with MIC and 52% with ITC received adjuvant therapy. Precise indication for adjuvant treatment was not specified in the study, and the adjuvant treatment was administered according to the internal guidelines of the individual institutions participating in the study. However, extrapolation of the data reported in the paper suggests that in a non-negligible proportion of cases, an indication for adjuvant treatment could have been set based on local risk factors. For instance, parametrial involvement was found in 7.1% of patients and represents per se an indication for adjuvant radio-chemotherapy irrespective of nodal status. Additionally, MIC were found in 12.7% of patients with FIGO IIA-IIB and 12.1% of patients with FIGO IB2 tumors. Considering that 36.2% and 43.6% of patients with IB2 and IIA-B tumors had LVSI in their series, it is easy to imagine that the majority of these patients would have received adjuvant treatment irrespective of nodal status. Although the presence of MIC was an independent prognostic factor for OS in the multivariate analysis, patients with locally advanced cervical cancer have independent poor prognostic factors. Therefore, the prognostic significance of LVM is challenging to assess in such a heterogeneous cohort of patients. From another perspective, the knowledge of nodal status in those patients requiring adjuvant treatment according to local tumor risk factors would have been superfluous since the treatment strategy would not have changed according to nodal status. This is in line with the role of LVM observed in endometrial cancer by Ghoniem et al. [26], who reported a higher risk of recurrence in patients with LVM in the SLN and uterine “high-risk” criteria, and an almost negligible risk of relapse in patients with low-risk early-stage endometrial cancer with ITC in the SLN.

Currently, only one series excluding patients who received adjuvant treatment is available. Colturato et al. [17] performed a retrospective ultrastaging pathologic analysis of pelvic nodes retrieved from patients with stage IB1-IIA cervical cancer who received exclusive surgical treatment (radical hysterectomy with bilateral PLND). Patients with LVM in the pelvic nodes had an 11.73 times higher risk of recurrence compared to node-negative patients. However, the study’s retrospective nature, the limited sample size, the posterior performance of ultrastaging, and the lack of information about the mean number of positive lymph nodes per patient make the results of this study unlikely to be generalizable. Similarly, Fregnani et al. [18] retrospectively evaluated nodal status by IHC pathologic analysis of pelvic lymph nodes in 289 patients with FIGO IB-IIA cervical carcinoma. In their series, only 36.4% of patients with LVM received adjuvant treatment, leading to a significantly worse prognosis of LVM patients compared to patients without LN metastases (HR 3.2; 95% CI: 1.1–9.6). Again, ultrastaging was performed posteriorly, information about the mean number of positive lymph nodes per patient and the site of recurrence was lacking, and patients with high-risk features were included in the analysis. For instance, 11% of patients had a tumor greater than 4 cm in size, 2.5% had parametrial involvement, and 4% had positive surgical margins, which are per se associated with a higher risk of tumor recurrence.

Despite the limitations, the findings of these retrospective series have led to the routine administration of adjuvant therapy in patients with LVM, especially if MIC are detected. The change over time in clinicians’ attitudes to treating LVM is highlighted by the two French prospective trials SENTICOL 1 and SENTICOL 2 [3,4,11]. In contrast with SENTICOL 2 [4], where patients with both MIC and ITC were treated with adjuvant radio-chemotherapy, SENTICOL 1 [3] was designed when SLN biopsy had not been implemented in routine clinical practice and the issue of LVM had not emerged yet. Patients with LVM received adjuvant treatment if other concomitant risk factors were present, whereas an isolated finding of MIC or ITC did not represent an indication for further treatment. Consequently, in SENTICOL 2, a higher rate of adjuvant treatment was recorded compared to SENTICOL 1. Despite this, no statistical difference in DFS was found between patients with LVM receiving adjuvant radio-chemotherapy compared to those who did not (1/13 (7.7%) vs. 1/11 (9%), *p* = NA) [11]. Of note, the population included in both studies represents those patients with a true primary surgical indication, since only those with a tumor size of less than 4 cm were included in both studies. However, some other aspects must be considered. First, the small sample size and the low numbers of events recorded in both trials make these studies underpowered for detecting any survival impact of LVM. Second, in SENTICOL 1, all patients received concurrent SLN biopsy and systematic PLND. This surgical approach could have mitigated the lower use of adjuvant radio-chemotherapy in SENTICOL 1 patients. In fact, in the combined SENTICOL 1 and 2 study by Guani et al. [11], the authors performed a DFS analysis stratifying patients according to the type of nodal staging they received, studying patients with SLN biopsy alone versus those with SLN biopsy followed by PLND. The authors found a favorable DFS trend for patients with LVM treated with concomitant SLN biopsy and PLND compared to those receiving SLN biopsy alone. One out of 6 patients receiving SLN biopsy alone experienced recurrence, whereas 1 out of 17 patients receiving SLN biopsy plus PLND had recurrence. Although of no statistical significance, this trend seems to confirm the findings of Zaal et al. [27], who performed a multicenter retrospective analysis to evaluate the impact of PLND following SLN biopsy in 645 patients with FIGO IA-IIB cervical cancer. PLND was found to give no survival advantage in patients with negative or macrometastatic SLN. In contrast, an overall survival benefit was described if a systematic PLND with the removal of more than 16 nodes was performed in patients with LVM in the SLN.

### 3.3. Impact of the Site of Recurrence: Is It Negligible?

The results of the LACC trial by Ramirez et al. [28] have elucidated a negative impact on DFS and OS of minimally invasive surgery compared to laparotomy and have led to a drastic change worldwide in the surgical approach for cervical cancer treatment. Most of the data available on the oncologic impact of LVM refer to the pre-LACC era, when surgeons’ preferences and skills guided the surgical approach. Although information about the surgical approach is not retrievable in most of the studies, a non-negligible number of patients received minimally invasive surgery, which could have impacted the incidence and pattern of recurrences. For example, among the 321 patients in the combined SENTICOL 1 and 2 population, 217 underwent a laparoscopic procedure, whereas only 21 received open surgery [11]. Similarly, 46% of patients included in Buda et al.’s series [13] underwent laparoscopic radical hysterectomy.

Even if it is of great importance in the definition of the prognostic relevance of LVM and the decision of the subsequent lines of treatment, information on recurrence sites (central pelvic, lateral pelvic, peritoneal, or even intra- or extra-abdominal metastatic) is almost always lacking for patients in most of the studies previously reported [29,30,31]. For the only patient with MIC in the SLN who developed recurrence in the SENTICOL 1 trial, recurrent disease was recorded as pelvic with no further distinction between central parametrial and pelvic side wall recurrence. Similarly, among the patients with LVM who recurred in the series by Kocian et al. [16], no precise definition of the site of recurrence was specified. Two patients developed distant metastatic disease, one developed a recurrence in the pelvis, and three patients developed both distant and local recurrent disease. No distinction between distant parenchymal and/or para-aortic nodal involvement nor between parametrial and/or pelvic wall recurrence was specified. It is our opinion that knowledge about the site of recurrence of patients with LVM is of great importance since the true impact of having LVM on prognosis and its weight among other risk factors would be elucidated, even after adjuvant treatment administration. For instance, in the small prospective series by Nica et al. [24], the only patient treated with a radical hysterectomy who had LVM in the SLN and experienced central pelvic recurrence presented a high-risk disease, with parametrial node involvement and positive vaginal margins. We think that this patient would have recurred irrespective of SLN micrometastatic involvement. Again, the coexistence of multiple high-risk factors makes it impossible to assess the true impact of LVM on prognosis.
cancers-16-01107-t001_Table 1Table 1Exploration of the available literature: study design, population, and outcomes.AuthorDateStudy DesignPopulationFIGO StagePrevalence of Isolated LVM Surgical LN TreatmentAdjuvantTreatmentNegative Impact of LVM on Oncologic Outcome *LVMITC OnlyMIC OnlyNode-NegativeLVMEndpointLVMMICITCMarchiolè et al. [19]2005Retrospective52IA2-IIB (FIGO 1988)12/52 (23%)6/52 (11.5%)6/52 (11.5%)PLNDNANARRYesYesYesFregnani et al [18].2006Retrospective289IB-IIA (FIGO 1988)11/289 (3.8%)NANAPLNDNA4/11 (36%)DFSYesNANACibula et al. [15]2012Retrospective645IA1-IIB (FIGO 2009)71/645 (11%)25/645 (4%)46/645 (7%)SLN + PLND48/456 (10.5%)51/71 (72%)RFSOSNANAYesYesNoNoStany et al. [12]2015Retrospective129IA2-IB2 (FIGO 1988)26/129 (20%)NANAPLND19/103 (18.5%)10/26 (38%)RFS, OSNoNoNANAColturato et al. [17]2016Retrospective83IB1-IIA (FIGO 2009)6/83 (7%)NANAPLND0/77 (0%)0/6 (0%)RRYesNANAGuani et al. [20]2019Prospective139IA1-IB1 (FIGO 2009)13/139 (9%)6/139 (4%)7/139 (5%)SLN + PLNDNA4/13 (30%)DFSNoNoNoNica et al. [24]2019Prospective19IA1-IB3 (FIGO 2018)NA9/19 (47%)10/19 (53%)SLN or SLN + PLNDNA14/19 (74%)RFSNoNoNoKocian et al. [16]2020Retrospective226IA1-IIB (FIGO 2009)24/226 (11%)8/226 (4%)16/226 (7%)SLN or SLN + PLNDNA17/24 (71%)DFSOSNANAYes YesNoNoBuda et al. [13]2020Retrospective573IA1-IB2 (FIGO 2018)21/573 (3.6%)4/573 (0.6%)17/573 (3%)SLN or SLN + PLNDNANADFSNoNoNoGuani et al. [11]2020Prospective321IA1-IB1 (FIGO 2009)24/321 (7%)24/321 (4%)11/321 (3%)SLN or SLN + PLNDNA13/24 (54%)DFSNoNoNoDostalek et al. [21]2023Retrospective967IA1-IIB (FIGO 2018)93/967 (10%)39/967 (4%)54/967 (6%)SLN or SLN + PLND151/795 (19%)71/93 (76%)DFSNAYesYesRR—Risk of Recurrence; DFS—Disease-Free Survival; RFS—Recurrence Free Survival; OS—Overall Survival; *—compared to negative nodal status.

## 4. Present and Future Challenges

The available literature results present some limitations about the definition of LVM, the role of ITC when considered separately from MIC, and the heterogeneity among the studies available. Therefore, some questions still need to be answered.

One of the first challenges would be understanding the actual prevalence of LVM in specific populations of patients based on the local risk factors, like stage and LVSI. This will help with understanding the real impact of the problem, especially in patients who will not be candidates for adjuvant treatment (e.g., low-risk and intermediate-risk populations).

In current 2023 ESGO-ESTRO-ESP guidelines [1], adjuvant radio-chemotherapy is indicated in cases of metastatic involvement of pelvic lymph nodes (either MAC or MIC) with a IV-A level of evidence, whereas it may be considered in case of ITC in the SLN with a IV-C level of evidence. These statements rely mainly on the results of retrospective studies with great heterogeneity in patient selection and adjuvant treatment indications. Particularly, a non-negligible number of patients included in the studies evaluating the impact of LVM received adjuvant treatment according to local risk factors, irrespective of the presence of LVM in the pelvic nodes, and this may have interfered with estimation of the real impact of LVM on prognosis. Adjuvant treatment in the management of early-stage cervical cancer is indicated in cases with high-risk criteria, namely parametrial infiltration, positive margins, and lymph node involvement, whereas its use in the case of Sedlis’ intermediate-risk factors [23] is a matter of debate and not universally accepted. If the results of the ongoing CERVANTES Trial [32] are found to not support the use of adjuvant treatment in cases with intermediate-risk factors, it is our opinion that the evaluation of patients with nodal LVM and no further local risk factors requiring adjuvant treatment will be the best chance for assessing the true impact of LVM on prognosis.

To our knowledge, the prognostic impact of LVM, either MIC or ITC, which is confined to the SLN (without further pelvic lymph node involvement) has not yet been investigated. Two ongoing trials, SENTIX [5] and SENTICOL 3 [6], will probably confirm the reliability of SLN biopsy alone for nodal staging in early-stage cervical cancer. They will likely align with the results of the metanalysis recently published by Parpinel et al. [33], showing similar survival outcomes after SLN biopsy alone and PLND. If SLN biopsy without concurrent PNLD is to become the standard surgical assessment of nodal status, the scientific community has to first evaluate the prognostic impact of LVM in the SLN without further nodal involvement. In many of the retrospective series cited above, the mean number of positive lymph nodes is not mentioned and, in some cases, SLN assessment was not even performed. In Kocian et al.’s series [16], a comprehensive description of nodal status was reported that combined the results of SLN ultrastaging and standard H&E analysis of non-sentinel pelvic lymph nodes. Specifically, 1 patient with ITC and 2 patients with MIC in their SLN had ITC and MIC in non-sentinel pelvic nodes, respectively, whereas 7 patients with ITC and 14 patients with MIC in the SLN had no further involvement of pelvic lymph nodes. However, no specific analysis was performed for patients with LVM confined to the SLN only. Recently, Pache et al. [34] published an ancillary analysis of SENTICOL 1-2 trials to exploit the risk factors associated with non-SLN metastases in patients with positive SLN. All non-SLNs were analyzed via ultrastaging to avoid missing LVM. Among 52 patients with at least one positive SLN, metastatic involvement of non-SLN was identified in 7 patients (13.5%). Elder age and LVSI were independently associated with non-SLN involvement. Interestingly, among the seven patients with non-SLN metastatic involvement, four patients had MAC in their SLN, one patient had ITC in the SLN and LVM (MIC + ITC) in non-sentinel lymph nodes, and two patients had MIC in their SLN and MAC non-sentinel lymph nodes.. Unfortunately, no survival analysis was carried out, and the prognostic impact of multiple LVM involvement of both the SLN and non-SLNs remains unclear. We believe that isolated LVM in the SLN only may have a better impact on prognosis than multiple nodal involvement. Further investigations should be carried out on this topic; we believe that the risk of further nodal involvement other than the SLN should be accurately assessed, especially if ITC are found. According to the data we presented, MIC are associated with significantly higher risks of recurrence and death compared to node-negative patients and seem to require adjuvant treatment. In contrast, no conclusive data are available on ITC. Probably, isolated MIC in the SLN would need further treatment equal to that for MAC or multiple-MIC patients; on the contrary, we think that adjuvant treatment in the case of ITC may be an over-treatment, especially if it is demonstrated that ITC in the SLN are unlikely to be associated with further nodal involvement.

Another critical challenge will be to standardize the ultrastaging protocols of the SLN worldwide. Different protocols may lead to varying prevalences of LVM in the same patient population. Additionally, the size and definition of MAC, MIC, and ITC are based on the AJCC recommendations on breast cancer and applied to cervical cancer as well as other gynecological cancers (i.e., endometrial and vulvar). However, validation for each gynecological cancer would be beneficial. This is an urgent need for all gynecological cancers in which SLN is now routine clinical practice (cervical, endometrial, and vulvar cancer).

## 5. Conclusions

SLN biopsy represents one of the most valuable achievements in the era of precision oncology. As for other malignancies, SLN biopsy has led to a higher diagnostic performance of nodal status assessment compared to standard bilateral PLND in the treatment of early-stage cervical cancer. Implementing SLN in routine clinical practice has raised the question of whether LVM are clinically relevant in treating cervical cancer. As for endometrial cancer, the risk of relapse and the need for adjuvant treatment depend on multiple risk factors, many of which are interlaced with each other and can be difficult to extricate. This makes the prognostic impact of LVM challenging to evaluate and the studies currently available present many limitations. Particularly, the literature is debating the impact of LVM on prognosis and whether adjuvant radio-chemotherapy is indicated in this group of patients. Consequently, based on the results of the recently published meta-analysis by Guani et al. [20], we believe that the prognoses of patients with micrometastatic pelvic lymph node involvement seem to be similar to those of patients with MAC. In contrast, no conclusive data are available on the clinical significance of ITC.

We believe that excluding confounders and an accurate selection of inclusion criteria to avoid the need for adjuvant treatment based on “local” risk factors would probably lead to a better understanding of the prevalence and prognostic impact of LVM in early-stage cervical cancer. Additionally, we believe that designing a prospective study to assess the impacts of LVM accurately would be challenging since cervical cancer is declining worldwide and early-stage cervical cancer has an excellent prognosis. The results of two prospective trials, SENTIX and SENTICOL 3, are awaited. We hope that accurate integration with available retrospective series will help to elucidate the prognostic roles of both MIC and ITC and to guide the choice of the most appropriate adjuvant treatment.

## Figures and Tables

**Figure 1 cancers-16-01107-f001:**
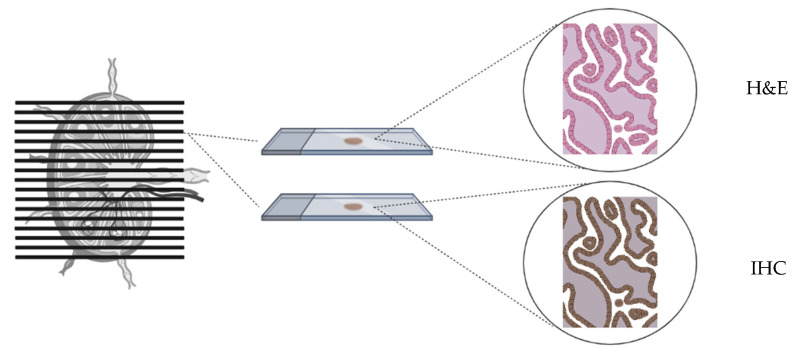
Pathologic ultrastaging of the SLN.

**Figure 2 cancers-16-01107-f002:**
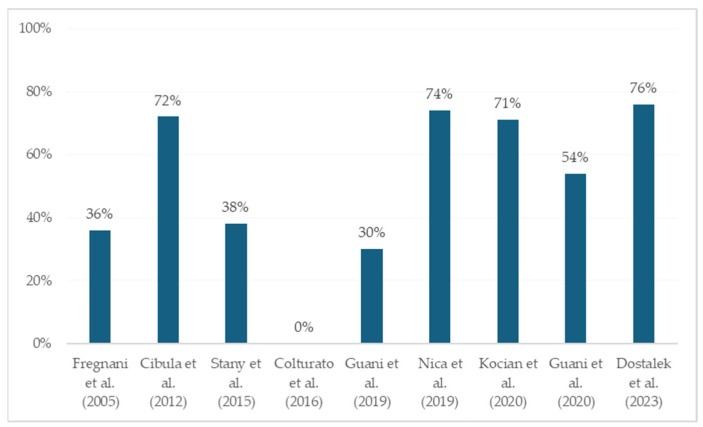
Rates of adjuvant treatment administration in patients with LVM [11,12,15,16,17,18,20,21,24].

## Data Availability

The data can be shared up on request.

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
