# Peer review of "Low-Volume Metastases in Cervical Cancer: Does Size Matter?"

_cancers, 2024, doi:10.3390/cancers16061107_

Round 1

Reviewer 1 Report

Comments and Suggestions for Authors

In this article, the authors present the definition, identification, prognostic and therapeutic role of low-volume metastases in early-stage cervical cancer and underline the present and future challenges. The manuscript is straightforward, well written, and concise and has clear results within the scope of a review article. Definitely deserves to be published and is a valuable contribution to the “cancersjournal. The following comments need to be addressed, as recommended.

[1] “3.3. The site of recurrence: is it negligible?”, Page 7 of 12, Lines 302-305:

Although information about the surgical approach is not retrievable in most of the studies, a non-negligible number of patients received minimally invasive surgery, which could have impacted the incidence and pattern of recurrences.”.

At that point, the authors should report that cervical cancer recurrences may be central pelvic, lateral pelvic and extra-pelvic. Central pelvic relapse can be located in the vaginal vault or usually involve the bladder and/or rectum. Lateral pelvic recurrence includes parietal and visceral pelvic side disease developed above and below the level of the obturator nerve, respectively.

Recommended reference: Boussios S, et al. Management of patients with recurrent/advanced cervical cancer beyond first line platinum regimens: Where do we stand? A literature review. Crit Rev Oncol Hematol. 2016;108:164-174.

[2] “3.3. The site of recurrence: is it negligible?”, Page 7 of 12, Lines 313-315:

Two patients developed distant metastatic disease, one developed a recurrence in the pelvis, and 3 patients developed both distant and local recurrent disease.”.

At that stage, the authors should mention that from the therapeutic perspective, there is increasing interest in the role of immunotherapy in advanced cervical cancer, particularly as the causative role of HPV is well established. A number of immunotherapy trials have been undertaken evaluating vaccine-based therapies, adoptive T-cell therapy and immune-modulating agents in patients with advanced cervical cancer.

Recommended reference: Choi S, et al. HPV and Cervical Cancer: A Review of Epidemiology and Screening Uptake in the UK. Pathogens 2023;12(2):298.

Comments on the Quality of English Language

Minor editing of English language required.

Author Response

Dear Reviewer,
Thank you for your precise and punctual response. We tried our best to improve our article following your suggestions. 

As per point (1), we reported the possible recurrent patterns of cervical cancer and included the reference you suggested, as long as two others about the topic. 

As per point (2) we thank you for the suggestions and carefully read the paper you indicated as reccomended reference. Although of great interest, we believe the topic of that article is not truly pertinent with the focus of our review and would not add a contibution to our paper. Nevertheless, we will take it into account for other ongoing works among ours. 

Thank you again for your reviews.

We now believe that our paper is a better insight into the current evidence about the role of LVM in early-stage cervical cancer.

We hope that our implementations have made the article suitable for publication.

Reviewer 2 Report

Comments and Suggestions for Authors

Low-volume metastases in cervical cancer: does size matter?

Manuscript ID: cancers-2911764

The implementation of sentinel lymph node (SLN) biopsy is changing the scenario in the surgical treatment of early-stage cervical cancer and the oncologic safety of replacing bilateral pelvic lymphadenectomy with SLN biopsy is currently under investigation. One of the  undisputed values of SLN biopsy is its diagnostic accuracy in detecting low-volume metastases (LVM) via pathologic ultrastaging. In early-stage cervical cancer, the reported incidence of LVM ranges from 4 to 20%. The prognostic impact and the role of adjuvant treatment in patients with LVM is still unclear. The authors have attempted to deep dive into highlighting the unanswered questions about the definition, identification, prognostic and therapeutic role of LVM and underline the present and future challenges we are facing.

The review article is well presented by the authors. All the major topics are discussed in dept in the review article. The article can be accepted in the current form with certain additions.

Minor comments:

The authors should incorporated more pictures/figures about the topic being discussed.

1. The authors should include more figures about the current topic.

Author Response

Dear Reviewer,
Thank you for your precise and punctual response. We tried our best to improve our article following your suggestions. 

We added Figure 2 to better visually describe the rate of adjuvant treatment among patients with LVM reported in the literature. 

Thank you again for your reviews.
We now believe that our paper is a better insight into the current evidence about the role of LVM in early-stage cervical cancer.

We hope that our implementations have made the article suitable for publication.